# An Effective Synchronization Approach to Stability Analysis for Chaotic Generalized Lotka–Volterra Biological Models Using Active and Parameter Identification Methods

**DOI:** 10.3390/e24040529

**Published:** 2022-04-09

**Authors:** Harindri Chaudhary, Ayub Khan, Uzma Nigar, Santosh Kaushik, Mohammad Sajid

**Affiliations:** 1Department of Mathematics, Jamia Millia Islamia, New Delhi 110025, India; hchaudhary@db.du.ac.in (H.C.); akhan12@jmi.ac.in (A.K.); uzmanigarkhan@gmail.com (U.N.); 2Department of Mathematics, Deshbandhu College, New Delhi 110019, India; 3Department of Mathematics, Bhagini Nivedita College, University of Delhi, New Delhi 110043, India; santosh.kaushik@bn.du.ac.in; 4Department of Mechanical Engineering, College of Engineering, Qassim University, Buraydah 51452, Saudi Arabia

**Keywords:** active control method, chaotic system, generalized Lotka–Volterra model, Lyapunov stability theory, parameter identification method, projective synchronization

## Abstract

In this manuscript, we systematically investigate projective difference synchronization between identical generalized Lotka–Volterra biological models of integer order using active control and parameter identification methods. We employ Lyapunov stability theory (LST) to construct the desired controllers, which ensures the global asymptotical convergence of a trajectory following synchronization errors. In addition, simulations were conducted in a MATLAB environment to illustrate the accuracy and efficiency of the proposed techniques. Exceptionally, both experimental and theoretical results are in excellent agreement. Comparative analysis between the considered strategy and previously published research findings is presented. Lastly, we describe an application of our considered combination difference synchronization in secure communication through numerical simulations.

## 1. Introduction

Preserving and addressing ecological or biological systems are primary concerns of many scientific areas. Consequently, their significant adverse consequences, for instance, the occurrence of extreme complex dynamic behavior in the above-mentioned systems due to oscillatory interactions found in the population through competition or cooperation are challenging topics for researchers, ecologists, biologists. Mathematical models give both pragmatic and quantitative descriptions of significant biological phenomena, and bioscience interpretations of their outcomes would help in practical predictions of the state of a considered system under different conditions. The concept of employing mathematical models for prey–predator interactions was independently introduced by A. J. Lotka [1] and V. Volterra [2] in the 1920s to examine many intriguing properties existing in population dynamics such as predation and parasitism. Subsequently, numerous mathematical models on prey and predator populations were presented and studied by several researchers and authors, resulting in extending the applicability of such models [3,4,5,6,7,8,9,10,11,12,13]. Moreover, system parameters have a prominent aspect in controlling or chaotifying considered chaotic models, and consequently in synchronization and control theory, thereby rendering parameter identification techniques a major key factor in chaos theory. More importantly, remarkable works [14,15,16,17] were reported in this field utilizing the design of the considered adaptive control method to estimate unknown parameters.

Specifically, the generalized Lotka–Volterra (GLV) biological model comprising three species is the most influential model in existing population interactions. Significantly, Arnedo et al. [18] in 1980 reported that it may acquire a chaotic pattern for a considerable set of parameters. These models essentially contain one prey and two predator populations. In addition, Samardzija and Greller [19] in 1988 comprehensively showed that GLV systems possess chaotic behavior. Chaotic systems are basically nonlinear dynamic systems with extreme sensitiveness to small perturbations of initial conditions and parameter data. Synchronization in chaotic systems is defined as the process of typically adapting chaotic systems, so that each shows similar behavior owing to coupling for stability gains.

It has been more than three decades since the pioneering announcement of the chaos phenomenon by Pecora and Carroll [20] in 1990. They systematically developed a synchronization process using a master–slave configuration in similar chaotic models with entirely different initial conditions. Afterwards, synchronization in nonidentical chaotic systems was also established. Since then, a huge range of newly prescribed chaos synchronization and control schemes have been initiated and analysed by researchers and academicians. Various synchronization techniques, such as complete [21,22], hybrid [23], anti [24], partial anti [25], projective [26], hybrid projective [17,27,28,29], function projective [30], phase [31], combination synchronization [32], lag [33], combination-combination [16,17], modified projective [34], compound [35], triple compound [36], combination difference [24,37], modulus synchronization [38,39], output-feedback chaos synchronization [40], partial synchronization [41] and multiclustering synchronization [42], in chaotic and hyperchaotic systems are attained by utilizing enormous control approaches, namely, active [43,44,45,46], adaptive [17,47,48], backstepping design [49], feedback [47], sliding mode [50], adaptive sliding mode control [51], and UDE-based control method [52], which are available in the recently updated literature. Moreover, synchronization theory for time-delayed nonlinear and fractional-order (FO) systems was precisely developed. Chaos control in chaotic systems by employing a parameter identification method was introduced by Hubler [53] in 1989. Further, E.W. Bai and K. E. Lonngren [54] achieved synchronization in chaotic systems via an active control method in 1997. More importantly, combination synchronization was first studied in 2011 by Runzi et al. [55]. Further, many significant studies [56,57] were conducted in this direction. In addition, Dongmo et al. [58] introduced difference synchronization in 2018. Optimal control design and synchronization for LV models were rigorously studied in [59]. Further, in [14,15], a parameter identification method was discussed in the synchronization of GLV biological systems.

Chaos synchronization has a huge spectrum of applications in secure communication [60,61,62,63,64,65,66,67]. Numerous types of secure communication strategies were illustrated, such as chaos modulation, chaos masking, and chaos shift keying. In chaos communication schemes, the essential idea of transmitting a message utilizing chaotic or hyperchaotic models is that a message signal is embedded in a transmitter system that generates a chaotic signal. After that, this chaotic signal is emitted to a receiver through a public channel. The message signal is lastly recovered by the receiver. A chaotic system is primarily used as both transmitter and receiver. Subsequently, this theory needs significant consideration in various research fields.

Our current paper’s objective, with the above works in mind, is to propose and analyze a combination difference projective synchronization (CDPS) technique in three identical chaotic GLV systems by utilizing active control and parameter identification methods. In combination difference synchronization schemes, three chaotic systems (identical or nonidentical) are involved, in which two are selected as master systems, and one is selected as a slave system. In this work, we considered the GLV model (master and slave system), but it is a nonrealistic mathematical model. Nevertheless, the mathematical aspect of the problem can shed some light on it.

The manuscript is organized as follows: Section 2 outlines the mathematical notations and basic terminology used within this paper. Section 3 presents a synchronization methodology in a general setup. Section 4 reports the chaotic analysis of GLV model for which CDPS was investigated. Active nonlinear controllers were appropriately designed for the CDPS scheme using Lyapunov stability theory. Section 5 describes CDPS via a parameter identification method (PIM), and discussions concerning the numerical simulations that were performed in MATLAB software are presented. Furthermore, comparative analysis with previously published findings was conducted. Section 6 comprehensively discusses an application of our considered approach, CDPS, in secure communication. Lastly, concluding remarks are in Section 7.

## 2. Problem Formulation

In this section, the methodology to elaborate combination synchronization [55] using master–slave composition in three chaotic systems is presented.

Let the first master system be
(1)$$\begin{array}{cc} {\dot{y}}_{m1}= & \;h_{1}\left(y_{m1}\right), \end{array}$$
and the second master system be
(2)$$\begin{array}{cc} {\dot{y}}_{m2}= & \;h_{2}\left(y_{m2}\right). \end{array}$$

Let the slave system be
(3)$$\begin{array}{cc} {\dot{y}}_{s1}= & \;h_{3}\left(y_{s1}\right)+U\left(y_{m1},y_{m2},y_{s1}\right), \end{array}$$
where $y_{m1}={\left(y_{m11},y_{m12},\ldots ,y_{m1n}\right)}^{T}\in R^{n}$, $y_{m2}={\left(y_{m21},y_{m22},\ldots ,y_{m2n}\right)}^{T}\in R^{n}$, $y_{s1}={\left(y_{s11},y_{s12},\ldots ,y_{s1n}\right)}^{T}\in R^{n}$ are state vectors of master and slave systems (1)–(3), respectively; $h_{1},h_{2},h_{3}:R^{n}\to R^{n}$ are three nonlinear continuous functions; and $U={\left(U_{1},U_{2},\ldots ,U_{n}\right)}^{T}:R^{n}\times R^{n}\times R^{n}\to R^{n}$ are controllers to be properly determined.

**Definition** **1.**
*Master Systems (1) and (2) are in complete synchronization (CS) with Slave System (3) if*

$$lim_{t\to \infty }∥y_{s1}\left(t\right)-\left(y_{m1}\left(t\right)+y_{m2}\left(t\right)\right)∥=0,$$

*where ∥.∥ denotes vector norm.*


**Definition** **2.**
*Master Systems (1) and (2) are in antiphase synchronization (APS) with Slave System (3) if*

$$lim_{t\to \infty }∥y_{s1}\left(t\right)+\left(y_{m1}\left(t\right)+y_{m2}\left(t\right)\right)∥=0.$$



**Definition** **3.**
*The combination of Master Systems (1) and (2) is in combination difference synchronization (CDS) with Slave System (3) if*

$$lim_{t\to \infty }∥e\left(t\right)∥=lim_{t\to \infty }∥Py_{s1}\left(t\right)-\left(Ry_{m2}\left(t\right)-Sy_{m1}\left(t\right)\right)∥=0,$$

*where ∥.∥ denotes vector norm and $P=diag\left(p_{1},p_{2},\ldots ,p_{n}\right),R=diag\left(r_{1},r_{2},\ldots ,r_{n}\right),S=diag\left(s_{1},s_{2},\ldots ,s_{n}\right)$ and P≠0.*


**Remark** **1.**
*Considered matrices P,R, and S are called scaling matrices. Moreover, P,R and S are expanded as matrices of functions of state variables $y_{m1},y_{m2}$ and $y_{s1}$.*


**Remark** **2.**
*The problem regarding combination synchronization is converted into a traditional chaos control issue [68] if R=S=0.*


**Remark** **3.**
*If P=I and R=S=βI, then for β=1, it is reduced to complete synchronization; if β=−1, it turns into antiphase synchronization. Hence, the combination difference projective synchronization (CDPS) error takes the form:*

(4)
$$\begin{array}{c} e\left(t\right)=y_{s1}\left(t\right)-\beta \left(y_{m2}\left(t\right)-y_{m1}\left(t\right)\right), \end{array}$$

*where β=diag(a,a,…,a).*


**Definition** **4.**
*The combination of Chaotic Systems (1)–(2) is in combination difference projective synchronization (CDPS) with System (3) if*

$$lim_{t\to \infty }∥e\left(t\right)∥=lim_{t\to \infty }∥y_{s1}\left(t\right)-\beta \left(y_{m2}\left(t\right)-y_{m1}\left(t\right)\right)∥=0.$$



The following section presents the general theory of the CDS scheme to control chaos generated by Chaotic Systems (1)–(3) using active control approach.

## 3. Synchronization Methodology

We now describe the methodology to achieve the CDS scheme between Master Systems (1) and (2), and Slave System (3). We designed controllers $U_{i}$ by
(5)$$\begin{array}{cc} U_{i}= & \;\frac{\theta_{i}}{p_{i}}-{\left(h_{3}\right)}_{i}-\frac{K_{i}e_{i}}{p_{i}}, \end{array}$$
where $\theta_{i}=\left(r_{i}{\left(h_{2}\right)}_{i}-s_{i}{\left(h_{1}\right)}_{i}\right)$ and $K_{i}>0$ (known as gain constants), i=1,2,3,…,n.

**Theorem** **1.**
*Considered Systems (1)–(3) asymptotically attain the proposed CDS scheme if controllers are defined as given in Equation (Equation 5).*


**Proof.** Errors are given by
$$\begin{array}{cc} e_{i}= & \;p_{i}y_{s1i}-\left(r_{i}y_{m2i}-s_{i}y_{m1i}\right),for\;i=1,2,\ldots ,n. \end{array}$$The error dynamic system turns into
(6)$$\begin{array}{cc} \dot{e_{i}}= & \;p_{i}{\dot{y}}_{s1i}-\left(r_{i}{\dot{y}}_{m2i}-s_{i}{\dot{y}}_{m1i}\right)=\;p_{i}\left({\left(h_{3}\right)}_{i}+U_{i}\right)-\left(r_{i}{\left(h_{2}\right)}_{i}-s_{i}{\left(h_{1}\right)}_{i}\right)\\ = & \;p_{i}\left({\left(h_{3}\right)}_{i}+\frac{\theta_{i}}{p_{i}}-{\left(h_{3}\right)}_{i}-\frac{K_{i}e_{i}}{p_{i}}\right)-\theta_{i}\\ = & \;-K_{i}e_{i}. \end{array}$$The classical Lyapunov function is defined by
(7)$$\begin{array}{cc} V(e(t))= & \;\frac{1}{2}e^{T}e\\ = & \;\frac{1}{2}\Sigma e_{i}^{2}. \end{array}$$On differentiating V(e(t)) as given in Equation (Equation 7), we have
(8)$$\begin{array}{cc} \dot{V}\left(e\left(t\right)\right)= & \;\Sigma e_{i}{\dot{e}}_{i}=\;\Sigma e_{i}\left(-K_{i}e_{i}\right)\\ = & \;-\Sigma K_{i}e_{i}^{2}),\;\mathrm{using}\;\mathrm{Equation}\;\left(6\right). \end{array}$$We now choose each $K_{i}>0\left(i=1,2,3,\ldots ,n\right)$, so that $\dot{V}\left(e\left(t\right)\right)$ given by Equation (Equation 8) is negative definite. Thus, by LST [69], we obtain
$$lim_{t\to \infty }e_{i}\left(t\right)=0,\;\;\left(i=1,2,3\right).$$Therefore, Master Systems (1) and (2), and Slave System (3) achieved the desired CDS scheme. □

## 4. Combination Difference Projective Synchronization (CDPS) for Identical Chaotic GLV Systems via Active Control Method (ACM)

In this section, we first describe the widely known chaotic GLV three-species system to be chosen for a CDPS scheme using active control design. Samardzija and Greller [19], primarily in 1988, exhibited that GLV systems possess chaotic behavior. We now present the GLV model as the first master system:(9)$$\begin{array}{c} \left\{\begin{array}{cc} {\dot{y}}_{m11}= & y_{m11}-y_{m11}y_{m12}+b_{3}y_{m11}^{2}-b_{1}y_{m11}^{2}y_{m13},\\ {\dot{y}}_{m12}= & -y_{m12}+y_{m11}y_{m12},\\ {\dot{y}}_{m13}= & b_{2}y_{m13}+b_{1}y_{m11}^{2}y_{m13}, \end{array}\right. \end{array}$$
where ${\left(y_{m11},y_{m12},y_{m13}\right)}^{T}\in R^{3}$ is the state vector of the system, and $b_{1}$, $b_{2}$ and $b_{3}$ are positive parameters. Additionally, in Equation (Equation 9), $y_{m11}$ denotes the prey population, and $y_{m12}$, $y_{m13}$ represents the predator populations. For parameter values $b_{1}=2.9851$, $b_{2}=3$, $b_{3}=2$ and initial values (27.5,23.1,11.4), the first master GLV system depicted chaotic behavior, as displayed in Figure 1a.

The second identical master GLV chaotic system is described as
(10)$$\begin{array}{c} \left\{\begin{array}{cc} {\dot{y}}_{m21}= & y_{m21}-y_{m21}y_{m22}+b_{3}y_{m21}^{2}-b_{1}y_{m21}^{2}y_{m23},\\ {\dot{y}}_{m22}= & -y_{m22}+y_{m21}y_{m22},\\ {\dot{y}}_{m23}= & b_{2}y_{m23}+b_{1}y_{m21}^{2}y_{m23}, \end{array}\right. \end{array}$$
where ${\left(y_{m21},y_{m22},y_{m23}\right)}^{T}\in R^{3}$ is the state vector of the system, and $b_{1}$, $b_{2}$ and $b_{3}$ are positive parameters. Further, in Equation (Equation 10), $y_{m11}$ represents the prey population, and $y_{m12}$, $y_{m13}$ denote the predator populations. For parameter values $b_{1}=2.9851$, $b_{2}=3$, $b_{3}=2$, this second master GLV system depicted chaotic behavior for selected initial conditions (1.2,1.2,1.2) as shown in Figure 1b.

The slave system, prescribed by the identical chaotic GLV system, is described as:(11)$$\begin{array}{c} \left\{\begin{array}{cc} {\dot{y}}_{s31}= & y_{s31}-y_{s31}y_{s32}+b_{3}y_{s31}^{2}-b_{1}y_{s31}^{2}y_{s33}+U_{1},\\ {\dot{y}}_{s32}= & -y_{s32}+y_{s31}y_{s32}+U_{2},\\ {\dot{y}}_{s33}= & b_{2}y_{s33}+b_{1}y_{s31}^{2}y_{s33}+U_{3}, \end{array}\right. \end{array}$$
where ${\left(y_{s11},y_{s12},y_{s13}\right)}^{T}\in R^{3}$ is the state vector of the system, and $b_{1}$, $b_{2}$ and $b_{3}$ are positive parameters. Moreover, in Equation (Equation 11), $y_{s31}$ represents the prey population, and $y_{s32}$, $y_{s33}$ denote the predator populations. For parameter values $b_{1}=2.9851$, $b_{2}=3$, $b_{3}=2$ and initial conditions (2.9,12.8,20.3), the slave GLV system displayed chaotic behavior, as exhibited in Figure 1c. Additionally, the detailed study and numerical results for Equations (9)–(11) are found in [19]. Further, $U_{1}$, $U_{2}$ and $U_{3}$ are controllers that are determined so that CDPS among identical GLV chaotic systems could be attained.

Next, CDPS is proposed to synchronize states of a chaotic GLV model. A Lyapunov stability theory (LST)-based active control approach was employed, and the required stability criterion is derived.

Synchronization error functions $\left(e_{1},e_{2},e_{3}\right)$ are defined as
(12)$$\begin{array}{c} \left\{\begin{array}{cc} e_{1}= & y_{s31}-\beta \left(y_{m21}-y_{m11}\right),\\ e_{2}= & y_{s32}-\beta \left(y_{m22}-y_{m12}\right),\\ e_{3}= & y_{s33}-\beta \left(y_{m23}-y_{m13}\right). \end{array}\right. \end{array}$$

The immediate goal here is the design of active controllers $U_{i},\left(i=1,2,3\right)$, which ensure that the synchronization error functions mentioned in Equation (Equation 12) satisfy
$$lim_{t\to \infty }e_{i}\left(t\right)=0,\;\;\left(i=1,2,3\right).$$

Then, the resulting error dynamics becomes
(13)$$\begin{array}{c} \left\{\begin{array}{cc} \dot{e_{1}}= & \;e_{1}-y_{s31}y_{s32}+b_{3}y_{s31}^{2}-b_{1}y_{s31}^{2}y_{s33}-\beta (-y_{m21}y_{m22}+b_{3}y_{m21}^{2}\\ & -b_{1}y_{m21}^{2}y_{m23}+y_{m11}y_{m12}-b_{3}y_{m11}^{2}+b_{1}y_{m11}^{2}y_{m13})+U_{1},\\ \dot{e_{2}}= & e_{2}+y_{s31}y_{s32}-\beta \left(-y_{m11}y_{m12}+y_{m21}y_{m22}\right)+U_{2},\\ \dot{e_{3}}= & b_{2}e_{3}+b_{1}y_{s31}^{2}y_{s33}-\beta \left(b_{1}y_{m21}^{2}y_{m23}-b_{1}y_{m11}^{2}y_{m13}\right)+U_{3}. \end{array}\right. \end{array}$$

Let us now design the active controllers by the following rule:(14)$$\begin{array}{cc} U_{1}= & \;\frac{\theta_{1}}{p_{1}}-{\left(h_{3}\right)}_{1}-\frac{K_{1}e_{1}}{p_{1}}, \end{array}$$
where $\theta_{1}=\left(r_{1}{\left(h_{2}\right)}_{1}-s_{1}{\left(h_{1}\right)}_{1}\right)$ and $K_{1}>0$ is a gain constant as described in Equation (Equation 5).

By inserting the values of $s_{1},r_{1},\theta_{1}$, ${\left(h_{3}\right)}_{1}$ into Equation (Equation 14) and simplifying, we obtain
(15)$$\begin{array}{cc} U_{1}= & \;-e_{1}+y_{s31}y_{s33}-b_{3}y_{s31}^{2}+b_{1}y_{s31}^{2}y_{s33}+\beta (-y_{m21}y_{m22}+b_{3}y_{m21}^{2}\\ & -b_{1}y_{m21}^{2}y_{m23}+y_{m11}y_{m12}-b_{3}y_{m11}^{2}+b_{1}y_{m11}^{2}y_{m13})-K_{1}e_{1}. \end{array}$$

Considering Equation (Equation 5), we obtain
(16)$$\begin{array}{cc} U_{2}= & \;\frac{\theta_{2}}{p_{2}}-{\left(h_{3}\right)}_{2}-\frac{K_{2}e_{2}}{p_{2}}, \end{array}$$
where $\theta_{2}=\left(r_{2}{\left(h_{2}\right)}_{2}-s_{2}{\left(h_{1}\right)}_{2}\right)$ and $K_{2}>0$ are gain constants.

By substituting the values of $s_{2},r_{2},\theta_{2}$, ${\left(h_{3}\right)}_{2}$ in Equation (Equation 16) and solving, we find that
(17)$$\begin{array}{cc} U_{2}= & \;e_{2}-y_{s31}y_{s32}+\beta \left(-y_{m11}y_{m12}+y_{m21}y_{m22}\right)-K_{2}e_{2}. \end{array}$$

Again using Equation (Equation 5), we have
(18)$$\begin{array}{cc} U_{3}= & \;\frac{\theta_{3}}{p_{3}}-{\left(h_{3}\right)}_{3}-\frac{K_{3}e_{3}}{p_{3}}, \end{array}$$
where $\theta_{3}=\left(r_{3}{\left(h_{2}\right)}_{3}-s_{3}{\left(h_{1}\right)}_{3}\right)$ and $K_{3}>0$ are gain constants.

By inserting the values of $s_{3},r_{3},\theta_{3}$, ${\left(h_{3}\right)}_{3}$ into Equation (Equation 18) and combining, we obtain
(19)$$\begin{array}{cc} U_{3}= & \;b_{2}e_{3}-b_{1}y_{s31}^{2}y_{s33}+\beta \left(b_{1}y_{m21}^{2}y_{m23}-b_{1}y_{m11}^{2}y_{m13}\right)-K_{3}e_{3}. \end{array}$$

On substituting the active controllers described in Equations (15), (17), and (19) into error dynamics Equation (Equation 13), we have
(20)$$\begin{array}{c} \left\{\begin{array}{cc} {\dot{e}}_{1}= & \;-K_{1}e_{1},\\ {\dot{e}}_{2}= & \;-K_{2}e_{2},\\ {\dot{e}}_{3}= & \;-K_{3}e_{3}. \end{array}\right. \end{array}$$

The Lyapunov function, denoted by V(e(t)), is now constructed using the following rule:(21)$$\begin{array}{cc} V(e(t))= & \;\frac{1}{2}\left[e_{1}^{2}+e_{2}^{2}+e_{3}^{2}\right]. \end{array}$$

Clearly, Lyapunov function V(e(t)), as defined in Equation (Equation 21), is surely positive definite in $R^{3}$. Then, the derivative for Lyapunov function is expressed as:(22)$$\begin{array}{cc} \dot{V}\left(e\left(t\right)\right)= & \;e_{1}{\dot{e}}_{1}+e_{2}{\dot{e}}_{2}+e_{3}{\dot{e}}_{3}. \end{array}$$

Using Equation (Equation 20) in Equation (Equation 22), one finds that
$$\begin{array}{cc} \dot{V}\left(e\left(t\right)\right)= & -K_{1}e_{1}^{2}-K_{2}e_{2}^{2}-K_{3}e_{3}^{2}\;<0, \end{array}$$
where $K_{i}>0$ for *i* = 1, 2, 3 which shows that $\dot{V}\left(e\left(t\right)\right)$ is surely negative definite. Therefore, by LST [69], CDPS error dynamics is globally asymptotical stable, i.e., synchronizing error function e(t)→0 is globally asymptotic for t→∞ for each initial values $e\left(0\right)\in R^{3}$.

## 5. Combination Difference Projective Synchronization (CDPS) in Identical Chaotic GLV Systems Using Parameter Identification Method (PIM)

In this section, we discuss the CDPS technique to obtain parameter-updating laws in order to identify and estimate system parameters, specifically in addition to adaptive controllers that all state variables tend to equilibrium points as time approaches infinity. As an illustrative example, we consider three identical GLV systems for investigating the CDPS scheme via PIM.

Two master systems (as GLV systems) and the slave system (as a GLV system) are written as follows:(23)$$\begin{array}{c} \left\{\begin{array}{cc} {\dot{y}}_{m11}= & y_{m11}-y_{m11}y_{m12}+b_{3}y_{m11}^{2}-b_{1}y_{m11}^{2}y_{m13},\\ {\dot{y}}_{m12}= & -y_{m12}+y_{m11}y_{m12},\\ {\dot{y}}_{m13}= & b_{2}y_{m13}+b_{1}y_{m11}^{2}y_{m13}. \end{array}\right. \end{array}$$
(24)$$\begin{array}{c} \left\{\begin{array}{cc} {\dot{y}}_{m21}= & y_{m21}-y_{m21}y_{m22}+b_{3}y_{m21}^{2}-b_{1}y_{m21}^{2}y_{m23},\\ {\dot{y}}_{m22}= & -y_{m22}+y_{m21}y_{m22},\\ {\dot{y}}_{m23}= & b_{2}y_{m23}+b_{1}y_{m21}^{2}y_{m23}. \end{array}\right. \end{array}$$
(25)$$\begin{array}{c} \left\{\begin{array}{cc} {\dot{y}}_{s31}= & y_{s31}-y_{s31}y_{s32}+b_{3}y_{s31}^{2}-b_{1}y_{s31}^{2}y_{s33}+W_{1},\\ {\dot{y}}_{s32}= & -y_{s32}+y_{s31}y_{s32}+W_{2},\\ {\dot{y}}_{s33}= & b_{2}y_{s33}+b_{1}y_{s31}^{2}y_{s33}+W_{3}, \end{array}\right. \end{array}$$
where $W_{1}$, $W_{2}$ and $W_{3}$ are control functions that were designed so that CDPS among three (identical) chaotic systems is obtained.

State errors are now defined as
(26)$$\begin{array}{c} \left\{\begin{array}{cc} e_{11}= & y_{s31}-\beta \left(y_{m21}-y_{m11}\right),\\ e_{12}= & y_{s32}-\beta \left(y_{m22}-y_{m12}\right),\\ e_{13}= & y_{s33}-\beta \left(y_{m23}-y_{m13}\right). \end{array}\right. \end{array}$$

The main goal of this considered work was to introduce controllers $W_{i},\left(i=1,2,3\right)$, ensuring that state errors defined in Equation (Equation 26) satisfied
$$lim_{t\to \infty }e_{1i}\left(t\right)=0,\;\;\left(i=1,2,3\right).$$

The subsequent error dynamic system is transformed as follows:(27)$$\begin{array}{c} \left\{\begin{array}{cc} \dot{e_{11}}= & e_{11}-y_{s31}y_{s32}+b_{3}y_{s31}^{2}-b_{1}y_{s31}^{2}y_{s33}-\beta (-y_{m21}y_{m22}\\ & +b_{3}y_{m21}^{2}-b_{1}y_{m21}^{2}y_{m23}+y_{m11}y_{m12}-b_{3}y_{m11}^{2}\\ & +b_{1}y_{m11}^{2}y_{m13})+W_{1},\\ \dot{e_{12}}= & -e_{12}+y_{s31}y_{s32}-\beta \left(-y_{m11}y_{m12}+y_{m21}y_{m22}\right)+W_{2},\\ \dot{e_{13}}= & -b_{2}e_{13}+b_{1}y_{s31}^{2}y_{s33}-\beta \left(b_{1}y_{m21}^{2}y_{m23}-b_{1}y_{m11}^{2}y_{m13}\right)+W_{3}. \end{array}\right. \end{array}$$

Now, we define adaptive control functions as follows:(28)$$\begin{array}{c} \left\{\begin{array}{cc} W_{1}= & -e_{11}+y_{s31}y_{s33}-\hat{b_{3}}y_{s31}^{2}+{\hat{b}}_{1}y_{s31}^{2}y_{s33}+\beta (-y_{m21}y_{m22}\\ & +{\hat{b}}_{3}y_{m21}^{2}-{\hat{b}}_{1}y_{m21}^{2}y_{m23}+y_{m11}y_{m12}-{\hat{b}}_{3}y_{m11}^{2}\\ & +{\hat{b}}_{1}y_{m11}^{2}y_{m13})-K_{1}e_{11},\\ W_{2}= & e_{12}-y_{s31}y_{s32}+\beta \left(-y_{m11}y_{m12}+y_{m21}y_{m22}\right)-K_{2}e_{12},\\ W_{3}= & {\hat{b}}_{2}e_{13}-{\hat{b}}_{1}y_{s31}^{2}y_{s33}+\beta \left({\hat{b}}_{1}y_{m21}^{2}y_{m23}-{\hat{b}}_{1}y_{m11}^{2}y_{m13}\right)-K_{3}e_{13}, \end{array}\right. \end{array}$$
where $K_{1},K_{2},K_{3}$ are gaining positive constants.

By inserting expressions for control functions described in Equation (Equation 28) into error dynamics Equation (Equation 27), we find that
(29)$$\begin{array}{c} \left\{\begin{array}{cc} {\dot{e}}_{11}= & \left(b_{3}-{\hat{b}}_{3}\right)y_{s31}^{2}-\left(b_{1}-{\hat{b}}_{1}\right)y_{s31}^{2}y_{s33}-\beta [\left(b_{3}-{\hat{b}}_{3}\right)y_{m21}^{2}\\ & -\left(b_{1}-{\hat{b}}_{1}\right)y_{m21}^{2}y_{s23}-\left(b_{3}-{\hat{b}}_{3}\right)y_{m11}^{2}\\ & -\left(b_{1}-{\hat{b}}_{1}\right)y_{m11}^{2}y_{m13}]-K_{1}e_{11},\\ {\dot{e}}_{12}= & -K_{2}e_{12},\\ {\dot{e}}_{13}= & -\left(b_{2}-{\hat{b}}_{2}\right)e_{13}+\left(b_{1}-{\hat{b}}_{1}\right)y_{s31}^{2}y_{s33}-\beta [-\left(b_{2}-{\hat{b}}_{2}\right)y_{m23}\\ & +\left(b_{1}-{\hat{b}}_{1}\right)y_{m21}^{2}y_{m23}+\left(b_{2}-{\hat{b}}_{2}\right)y_{m13}\\ & -\left(b_{1}-{\hat{b}}_{1}\right)y_{m11}^{2}y_{m13}]-K_{3}e_{13}, \end{array}\right. \end{array}$$
where ${\hat{b}}_{1}$, ${\hat{b}}_{2}$ and ${\hat{b}}_{3}$ are estimated values for unknown parameters $b_{1}$, $b_{2}$, and $b_{3}$, respectively.

Now, we define parameter estimation error by
(30)$$\begin{array}{c} {\tilde{b}}_{1}=b_{1}-{\hat{b}}_{1},{\tilde{b}}_{2}=b_{2}-{\hat{b}}_{2},{\tilde{b}}_{3}=b_{3}-{\hat{b}}_{3}. \end{array}$$

Using Equation (Equation 30), the error dynamics given in Equation (Equation 29) turns into
(31)$$\begin{array}{c} \left\{\begin{array}{cc} {\dot{e}}_{11}= & {\tilde{b}}_{3}\left(y_{s31}^{2}-\beta y_{m21}^{2}-\beta y_{m11}^{2}\right)-{\tilde{b}}_{1}(y_{s31}^{2}y_{s33}-\beta y_{m21}^{2}y_{m23}\\ & +\beta y_{m11}y_{m13})-K_{1}e_{11},\\ {\dot{e}}_{12}= & -K_{2}e_{12},\\ {\dot{e}}_{13}= & -{\tilde{b}}_{2}e_{13}+{\tilde{b}}_{1}\left(y_{s31}^{2}y_{s33}-\beta y_{m21}^{2}y_{m23}+\beta y_{m11}^{2}y_{m13}\right)-K_{3}e_{13}. \end{array}\right. \end{array}$$

The derivative of parameter estimation error, as defined in Equation (Equation 30), simplifies to
(32)$$\begin{array}{c} {\dot{\tilde{b}}}_{1}=-{\dot{\hat{b}}}_{1},{\dot{\tilde{b}}}_{2}=-{\dot{\hat{b}}}_{2},{\dot{\tilde{b}}}_{3}=-{\dot{\hat{b}}}_{3}. \end{array}$$

The Lyapunov function is described by
(33)$$\begin{array}{cc} V(e(t))= & \;\frac{1}{2}\left[e_{11}^{2}+e_{12}^{2}+e_{13}^{2}+{\tilde{b}}_{1}^{2}+{\tilde{b}}_{2}^{2}+{\tilde{b}}_{3}^{2}\right]. \end{array}$$

This clearly shows that Lyapunov function V(e(t)) is surely positive definite.

Using Equation (Equation 32), the derivative of Lyapunov function V(e(t)) becomes
(34)$$\begin{array}{cc} \dot{V}\left(e\left(t\right)\right)= & \;e_{11}{\dot{e}}_{11}+e_{12}{\dot{e}}_{12}+e_{13}{\dot{e}}_{13}-{\tilde{b}}_{1}{\dot{\hat{b}}}_{1}-{\tilde{b}}_{2}{\dot{\hat{b}}}_{2}-{\tilde{b}}_{3}{\dot{\hat{b}}}_{3}. \end{array}$$

Keeping Equation (Equation 34) in mind, we prescribe the parameter estimating laws by the following rule:(35)$$\begin{array}{c} \left\{\begin{array}{cc} {\dot{\hat{b}}}_{1}= & -\left(y_{s31}^{2}y_{s33}-\beta y_{m21}^{2}y_{m23}+\beta y_{m11}^{2}y_{m13}\right)E_{11}\\ & +\left(y_{s31}^{2}y_{s33}-\beta y_{m21}^{2}y_{m23}+\beta y_{m11}^{2}y_{m13}\right)e_{13}+K_{4}{\tilde{b}}_{1},\\ {\dot{\hat{b}}}_{2}= & -e_{13}^{2}+K_{5}{\tilde{b}}_{2},\\ {\dot{\hat{b}}}_{3}= & \left(y_{s31}^{2}+\beta y_{m11}^{2}-\beta y_{m21}^{2}\right)e_{11}+K_{6}{\tilde{b}}_{3}, \end{array}\right. \end{array}$$
where $K_{4},K_{5}$ and $K_{6}$ are gaining positive constants.

**Theorem** **2.**
*The considered chaotic Systems (9)–(11) asymptotically attained the proposed CDPS scheme in each initial state $\left(x_{m1}\left(0\right),x_{m2}\left(0\right),x_{m3}\left(0\right)\right)\in R^{3}$ if adaptive control functions and parameter estimating law were defined as given in Equations (28) and (35) respectively.*


**Proof.** It is obvious that V(e(t)), which is defined in Equation (Equation 33), is a positive definite Lyapunov function in $R^{6}$. By simplifying, Equations (31), (34) and (35) were transformed into the following expression:
$$\begin{array}{cc} \dot{V}\left(e\left(t\right)\right)= & \;-K_{1}e_{11}^{2}-K_{2}e_{12}^{2}-K_{3}e_{13}^{2}-K_{4}{\tilde{b}}_{1}^{2}-K_{5}{\tilde{b}}_{2}^{2}-K_{6}{\tilde{b}}_{3}^{2}\;<0, \end{array}$$
where $K_{i}>0$ for *i* = 1, 2, 3, 4, 5, 6. This shows that $\dot{V}\left(e\left(t\right)\right)$ is surely negative definite.Hence, by using LST [69], one can deduce discussed CDPS error e(t)→0 globally and asymptotically with t→∞ in each initial value $e\left(0\right)\in R^{3}$. □

### 5.1. Numerical Simulations and Results

Numerical simulations are specifically presented through MATLAB software to show the effectiveness of the CDPS scheme via ACM. We take here $p_{i}=1$ and $r_{i}=s_{i}=a=-2$ for all i=1,2,3, which shows that the considered slave model would be projectively antiphase synchronized with the combination of the given master systems. Further, $\left(K_{1},K_{2},K_{3}\right)$ were chosen to be 6. The initial conditions of Systems (9) and (10) and corresponding Slave System (11) were (27.5,23.1,11.4), (1.2,1.2,1.2), and (2.9,12.8,20.3), respectively. The trajectories of Master Systems (9) and (10), and Slave System (11) achieving projective antiphase synchronization are shown in Figure 2a–c. In addition, synchronization error functions $\left(e_{1},e_{2},e_{3}\right)=\left(-49.7,-31,-0.1\right)$ converging to zero for *t* tended to infinity, as shown in Figure 2d. Consequently, the discussed CDPS approach in master and slave systems is numerically demonstrated. Figure 3a–d exhibit the trajectories for Master Systems (9) and (10), and Slave System (11), attaining projective complete synchronization by choosing $p_{i}=1$, $r_{i}=s_{i}=\beta =1.5$ for all i=1,2,3 and $\left(e_{1},e_{2},e_{3}\right)=\left(42.35,45.65,35.6\right)$.

Further, numerical simulations were implemented through MATLAB software to show the effectiveness of the CDPS scheme via PIM. We take here $p_{i}=1$ and $r_{i}=s_{i}=a=1.5$ for all i=1,2,3, displaying that the considered slave chaotic system would be projectively completely synchronized with the combination of the given master systems. Further, $\left(K_{1},K_{2},K_{3}\right)$ were chosen to be 6. Initial conditions for Master Systems (9) and (10), and corresponding Slave System (11) were (27.5,23.1,11.4), (1.2,1.2,1.2), and (2.9,12.8,20.3), respectively. Trajectories for Master Systems (9) and (10), and Slave System (11) achieving projective complete synchronization are shown in Figure 4a–c. Furthermore, Figure 4d depicts that the estimated values $\left({\hat{b}}_{1},{\hat{b}}_{2},{\hat{b}}_{3}\right)$ of unknown parameters asymptotically converged to their originally described expressions with time. In addition, synchronization error functions $\left(e_{1},e_{2},e_{3}\right)=\left(-49.7,-31,-0.1\right)$ converging to zero for *t* tended to infinity, as shown in Figure 4e. As above, Figure 5a–e exhibit trajectories for Master Systems (9) and (10), and Slave System (11), attaining projective complete synchronization by choosing $p_{i}=1$, $r_{i}=s_{i}=\beta =-2$ for all i=1,2,3 and $\left(e_{1},e_{2},e_{3}\right)=\left(-49.7,-31,-0.1\right)$. Thus, the discussed CDPS scheme for master and slave systems was computationally confirmed.

### 5.2. Comparative Analysis

In [55], the authors initiated and achieved combination synchronization among 3 integer-order chaotic systems via an active backstepping method at *t* = 4 (approx.). In [56], the authors investigated an active backstepping method for achieving combination synchronization in integer-order chaotic systems, where synchronized states occurred at *t* = 4.5 (approx.). The researchers attained a finite-time stochastic combination synchronization scheme in 3 integer-order chaotic systems utilizing an adaptive method and the Weiner process in [57] at *t* = 3 (approx.). In [58], the researchers first proposed and discussed combination difference synchronization in 3 identical and nonidentical integer-order chaotic and hyperchaotic systems, where it was observed that synchronized states were realized at *t* = 6 (approx.). Moreover, the researchers in [70] discussed a feedback control strategy for achieving combination difference synchronization in three integer-order chaotic models comprising an exponential term at *t* = 4 (approx.). In addition, the hybrid synchronization of two chaotic systems was achieved via PIM in [15] when it was conducted on a similar GLV system with the same parametric values. Synchronized error converged to zero for *t* = 0.8 (approx.); in our study, the CDPS approach was attained by utilizing an active control approach and parameter identification method, in which synchronized errors converged to zero at *t* = 0.5 (approx.) and at *t* = 0.4 (approx.), respectively, as exhibited in Figure 6 and Figure 7. This obviously illustrates that our proposed CDPS approach utilizing an active control approach and parameter identification method is preferable to previous published work. Hence, synchronization time via our studied methodology was the least among all the above-discussed approaches, as shown in Table 1.

## 6. Application of Combination Difference Projective Synchronization in Secure Communication

In this section, we show the application of CDPS among GLVs. A chaotic signal is applied for message-masking and -recovery signals. The system block diagram of the GLV-based secure communication scheme is displayed in Figure 8. In a chaotic masking signal, information messaging signal Θ(t) is added at the master (transmitter) and slave (receiver) ends, and the message masking signal is removed. This application is based on the vast complexity of master systems to develop data security. Therefore, we divided the transmitted signals into two master systems to improve the protection of secure communication. Signals that receivers must receive are in the form of $\Theta \left(t\right)=\Theta_{1}\left(t\right)+\Theta_{2}\left(t\right)$, depicted in Figure 9a. Signals $\Theta_{1}\left(t\right)$ and $\Theta_{2}\left(t\right)$ are summed to the right-hand side of the third equation of the master systems. The amplitude of the message signal may be weaker than the chaotic masking signal, so that it cannot damage the chaotic system’s behavior. $\eta \left(t\right)=\Theta +\beta_{2}\left(y_{22m}-y_{12m}\right)$ is the transmitted signal shown in Figure 9b. Recovered signal $\hat{\Theta }\left(t\right)$ is obtained when the chaotic signal is subtracted from e(t), i.e., $\hat{\Theta }\left(t\right)=\eta \left(t\right)-y_{22s}$ exhibited in Figure 9c, and $\Theta \left(t\right)-\hat{\Theta }\left(t\right)$ demonstrates the error message signal in Figure 9d. We selected the signal to be $\Theta_{1}\left(t\right)=sign\left(sin\left(2*t\right)\right)$, $\Theta_{2}\left(t\right)=3*sign\left(sin\left(2*t\right)\right)$. Moreover, Figure 9a–d depict that message signal Θ(t)=2∗sign(sin(2∗t)) was successfully recovered at the receiver end.

## 7. Discussion and Conclusions

In this paper, a suggested CDPS strategy for chaotic identical GLV systems via active control and a parameter identification method was explored. By designing appropriate nonlinear controllers on the basis of classical LST, the considered CDPS scheme was attained. Additionally, special cases of antiphase synchronization, chaos control problem, and complete synchronization were discussed. Further, numerical simulations conducted in MATLAB exhibited that properly designed control functions are simple and efficient in asymptotically stabilizing the chaotic regime of GLV systems to the desired set points, which shows the effectiveness of the technique. Analytical and computational outcomes completely agreed. Comparison analysis showed that the time taken by synchronizing the error functions for converging to zero with time tending to infinity was less compared to that in other studies. This demonstrates that our considered CDPS design is more beneficial than earlier published work is, and our results indicate novelty over existing results. The discussed CDPS scheme has potential and advantages since this technique has enormous applications in encryption, control theory, and secure communication. In fact, we described the application of our considered CDPS in secure communication using chaos masking methodology. The considered scheme may be used to describe the effect of various specific coexisting species presented by the slave system of the GLV model. Controlling and examining chaos generated in the complex GLV systems of complex dynamic networks are open research problems. Thus, the investigated ACM and PIM methodologies can be developed for complex dynamical networks of the discussed GLV model as a future research problem.

## Figures and Tables

**Figure 1 entropy-24-00529-f001:**
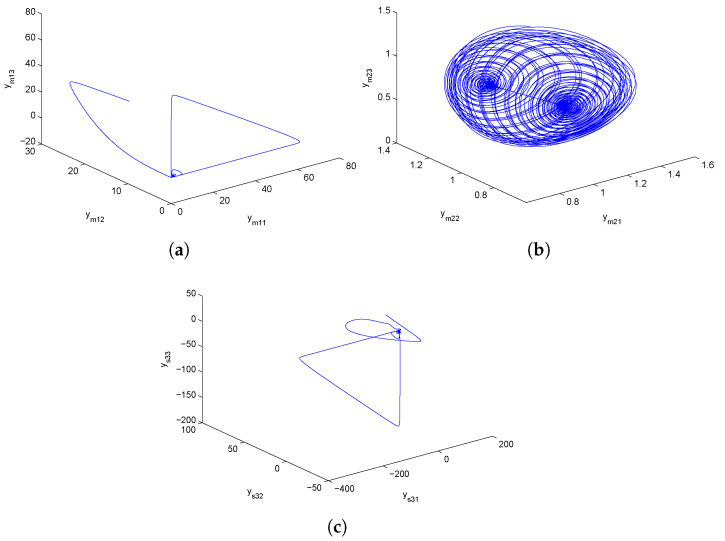
Phase plots for chaotic GLV system (**a**) $y_{m11}-y_{m12}-y_{m13}$ space, (**b**) $y_{m21}-y_{m22}-y_{m23}$ space, (**c**) $y_{s31}-y_{s32}-y_{s33}$ space.

**Figure 2 entropy-24-00529-f002:**
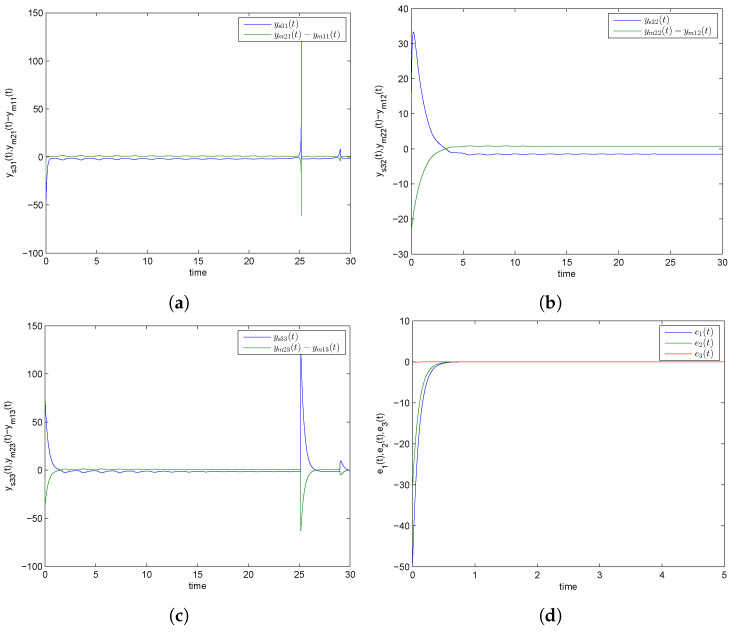
Time history of combination difference projective antiphase synchronized trajectories for GLV system (**a**) $y_{s31}\left(t\right)$ and $y_{m21}\left(t\right)-y_{m11}\left(t\right)$, (**b**) $y_{s32}\left(t\right)$ and $y_{m22}\left(t\right)-y_{m12}\left(t\right)$, (**c**) $y_{s33}\left(t\right)$ and $y_{m23}\left(t\right)-y_{m13}\left(t\right)$, (**d**) synchronization error plot.

**Figure 3 entropy-24-00529-f003:**
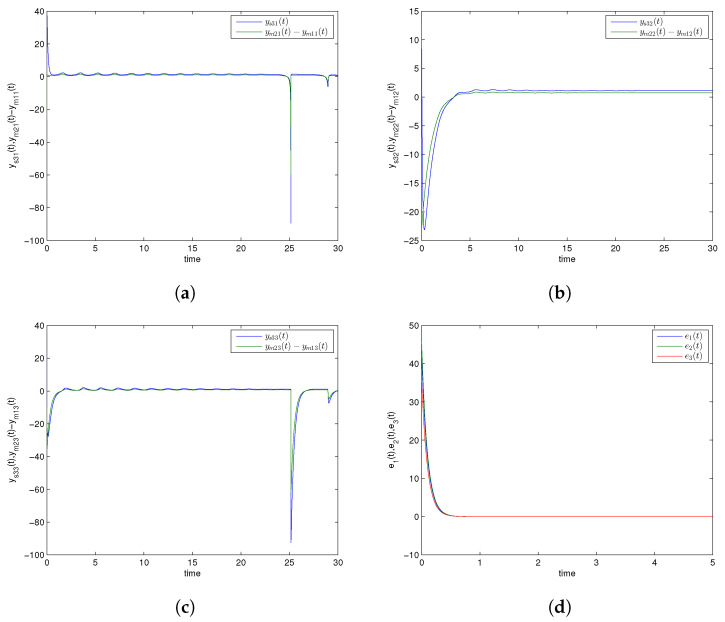
Time history of combination difference projective complete synchronized trajectories for GLV system (**a**) $y_{s31}\left(t\right)$ and $y_{m21}\left(t\right)-y_{m11}\left(t\right)$, (**b**) $y_{s32}\left(t\right)$ and $y_{m22}\left(t\right)-y_{m12}\left(t\right)$, (**c**) $y_{s33}\left(t\right)$ and $y_{m23}\left(t\right)-y_{m13}\left(t\right)$, (**d**) synchronization error plot.

**Figure 4 entropy-24-00529-f004:**
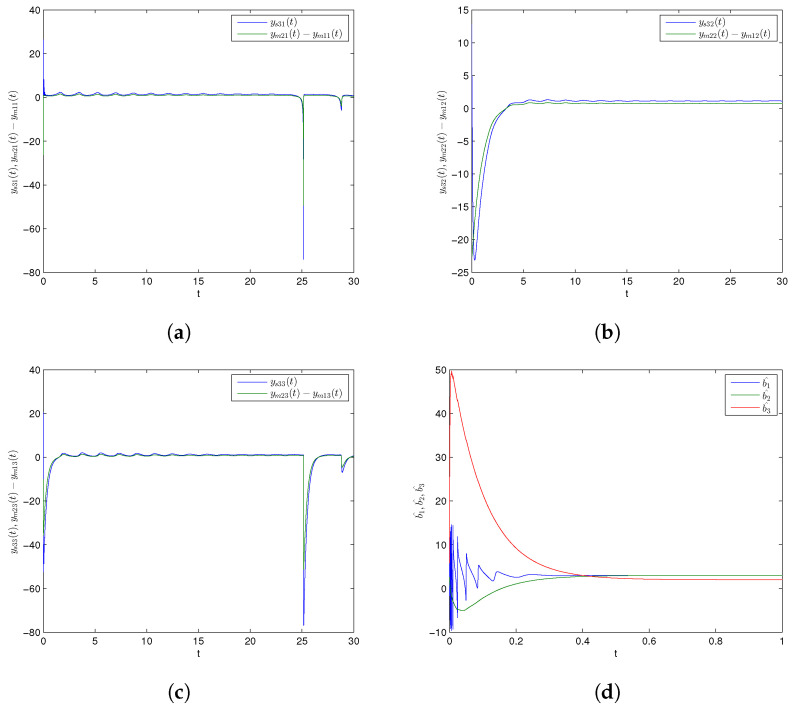
Time series for combination difference projective complete synchronized trajectories of GLV system (**a**) $y_{s31}\left(t\right)$ and $y_{m21}\left(t\right)-y_{m11}\left(t\right)$, (**b**) $y_{s32}\left(t\right)$ and $y_{m22}\left(t\right)-y_{m12}\left(t\right)$, (**c**) $y_{s33}\left(t\right)$ and $y_{m23}\left(t\right)-y_{m13}\left(t\right)$, (**d**) parameter estimation, (**e**) synchronization error plot.

**Figure 5 entropy-24-00529-f005:**
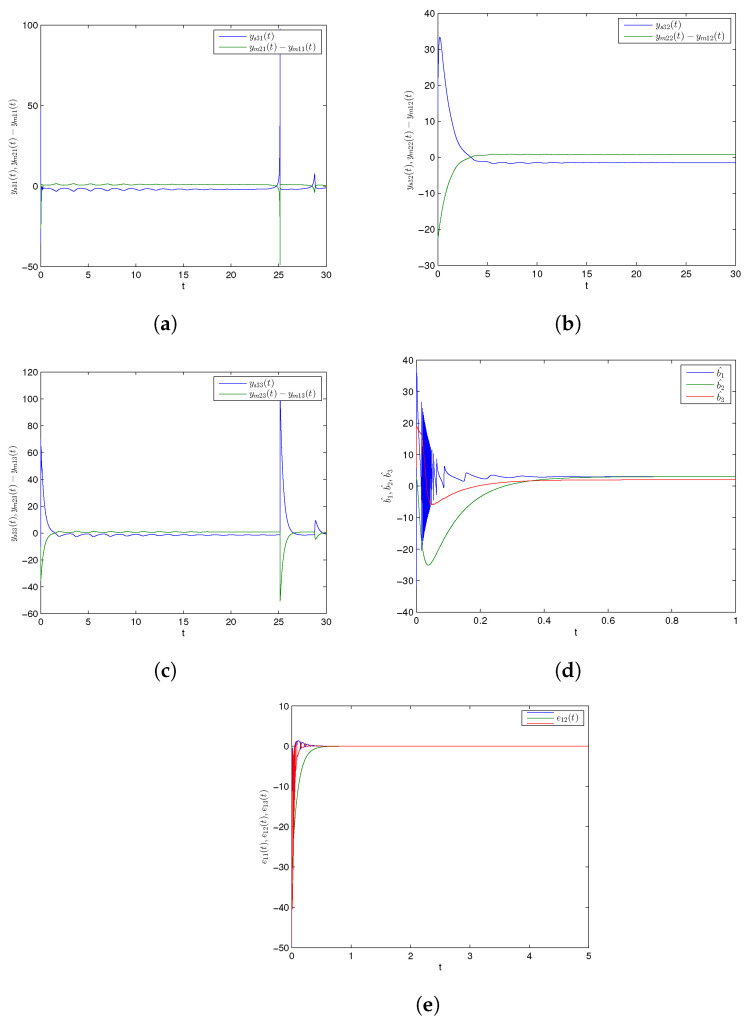
Time series for combination difference projective anti-phase synchronized trajectories of GLV system (**a**) $y_{s31}\left(t\right)$ and $y_{m21}\left(t\right)-y_{m11}\left(t\right)$, (**b**) $y_{s32}\left(t\right)$ and $y_{m22}\left(t\right)-y_{m12}\left(t\right)$, (**c**) $y_{s33}\left(t\right)$ and $y_{m23}\left(t\right)-y_{m13}\left(t\right)$, (**d**) parameter estimation, (**e**) synchronization error plot.

**Figure 6 entropy-24-00529-f006:**
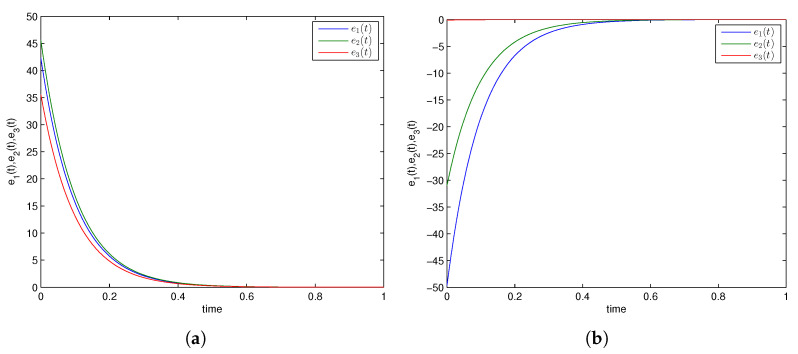
Time series of error convergence by active control method. (**a**) Combination difference projective complete synchronization; (**b**) combination difference projective antiphase synchronization.

**Figure 7 entropy-24-00529-f007:**
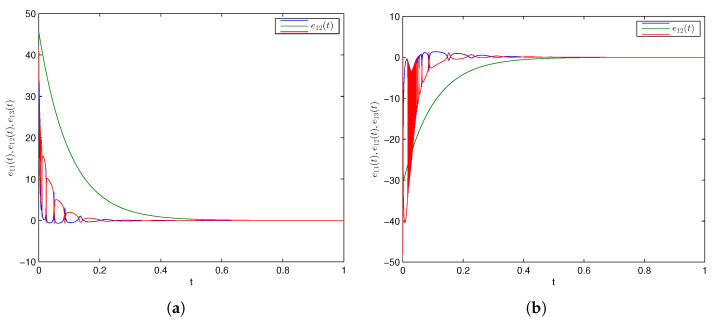
Time series of error convergence by parameter identification method. (**a**) Combination difference projective complete synchronization; (**b**) combination difference projective antiphase synchronization.

**Figure 8 entropy-24-00529-f008:**
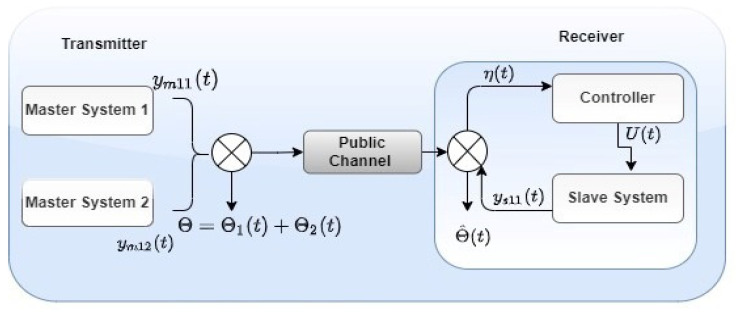
Combination difference synchronization-based secure communication.

**Figure 9 entropy-24-00529-f009:**
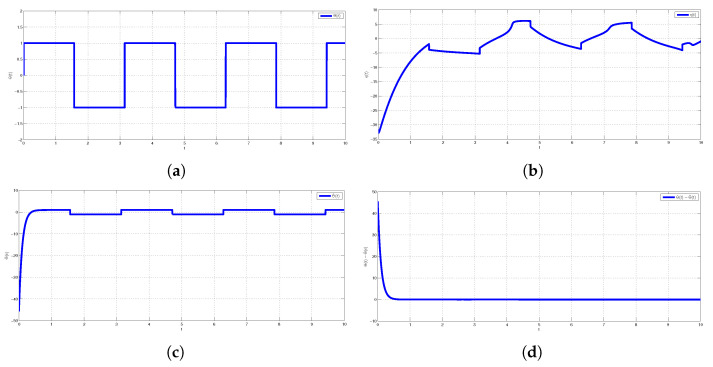
(**a**) Original message signal μ(t); (**b**) transmitted message signal e(t); (**c**) recovered signal $\hat{\mu }\left(t\right)$; (**d**) error message signal $\mu \left(t\right)-\hat{\mu }\left(t\right)$.

**Table 1 entropy-24-00529-t001:** Different types of synchronization schemes using different techniques.

Types of Synchronization	Authors	Time
1. Combination synchronization of three classical chaotic systems using active backstepping design	Runzi, Luo and Yinglan, Wang and Shucheng, Deng	4
2. Combination synchronization of three different order nonlinear systems using active backstepping design	Wu, Zhaoyan and Fu, Xinchu	4.5
3. Finite-time stochastic combination synchronization of three different chaotic systems and its application in secure communication	Runzi, Luo and Yinglan, Wang	3
4. Difference synchronization of identical and nonidentical chaotic and hyperchaotic systems of different orders using active backstepping design	Dongmo, Eric Donald and Ojo, Kayode Stephen and Woafo, Paul and Njah, Abdulahi Ndzi	6
5. Difference synchronization among three chaotic systems with exponential term and its chaos control	Yadav, Vijay K and Shukla, Vijay K and Das, Subir	4
6. Hybrid synchronization of generalized Lotka–Volterra three-species biological systems via adaptive control	Vaidyanathan, Sundarapandian	0.8
7. CDPS approach attained utilizing active control approach	Mohammad Sajid, Harindri Chaudhary, Ayub Khan, Uzma Nigar, Santosh Kaushik	0.5
8. CDPS approach attained using parameter identification method	Mohammad Sajid, Harindri Chaudhary, Ayub Khan, Uzma Nigar, Santosh Kaushik	0.4

## Data Availability

Not applicable.

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
