# Peer review of "An Effective Synchronization Approach to Stability Analysis for Chaotic Generalized Lotka–Volterra Biological Models Using Active and Parameter Identification Methods"

_entropy, 2022, doi:10.3390/e24040529_

Round 1

Reviewer 1 Report

The authors significantly revised the text and did a lot of useful corrections. Therefore, the present version of the manuscript looks better and clearer. Particularly, the term ‘anti-synchronization’ was changed to more appropriate ‘anti-phase synchronization’. At the same time, there are still some places where this notion wasn’t corrected: in captions of Fig.2, Fig. 5, Fig. 6, Fig. 7 and in the Discussion. Additionally, please check the writing of ‘Lotka-Volterra’ in Reference list. Finally, I didn’t find in the text any clarifications for the obtained negative values of the variables. What does it mean from the physical point of view? 

Author Response

Thank you very much for providing us an opportunity to revise the manuscript entitled “An Effective Synchronizational Approach to the Stability Analysis for Chaotic Generalized Lotka-Volterra Biological Models Using Active and Parameter Identification Methods” (ID 1600197).

We have endeavored to incorporate all possible revisions/corrections considering the Reviewers’/Editors’ comments, and the changes made in the text are highlighted in Red Colour. The point wise responses are enumerated herein below.

Comment 1: The authors significantly revised the text and did a lot of useful corrections. Therefore, the present version of the manuscript looks better and clearer. Particularly, the term ‘anti-synchronization’ was changed to more appropriate ‘anti-phase synchronization’. At the same time, there are still some places where this notion wasn’t corrected: in captions of Fig.2, Fig. 5, Fig. 6, Fig. 7 and in the Discussion.

Additionally, please check the writing of ‘Lotka-Volterra’ in Reference list. Finally, I didn’t find in the text any clarifications for the obtained negative values of the variables. What does it mean from the physical point of view? 

Response: I have checked throughout the paper very carefully and made changes in all the Figures 2, 5, 6 and 7 as per the suggestion given the learned reviewer.

Further, I have made corrections in the writing of ‘Lotka-Volterra’ in Reference list. Also, we have deleted the Figure 1 (c) (before revision) showing negative values of the variables as per the suggestion given by another learned Reviewer. In fact, Figure 1 (c) is a projection of the Figure 2 (c) (before revision) which is now Figure 1 (c) in the revised manuscript and here all the variables obtained positive values..

Reviewer 2 Report

The paper deals with a synchronization and control problem for two master and one slave systems of generalized Lotka-Volterra type. There are two quite fair approaches, first of them based on chosen controllers, while in the second method the choices are related to the variation of the parameters.

Despite the fact that, in both cases, the choices are made in a very simple and predictible maner, so that part of the terms from the master systems to be eliminated, the paper deserves to be published in its present form, with some minor revisions:

  • The authors have to check the third equation from (28) and the third equation in (29) and to correct the typing errors.
  • At the end of the paper (section 7) the authors should outline better the novelty they brings. The subsection 5.2. could be extended to include previous results both on ACM and on PIM and moved here.

Author Response

Thank you very much for providing us an opportunity to revise the manuscript entitled “An Effective Synchronizational Approach to the Stability Analysis for Chaotic Generalized Lotka-Volterra Biological Models Using Active and Parameter Identification Methods” (ID 1600197).

We have endeavored to incorporate all possible revisions/corrections considering the Reviewers’/Editors’ comments, and the changes made in the text are highlighted in Red Colour. The point wise responses are enumerated herein below.

Comment 1: The authors have to check the third equation from (28) and the third equation in (29) and to correct the typing errors.

Response: The authors are thankful to the learned reviewer for pointing out this typographical error. The third equation from 28 and 29 Eqs. have been replaced and marked as Red Colour in revised manuscript.

Comment 2: At the end of the paper (section 7) the authors should outline better the novelty they brings. The subsection 5.2. could be extended to include previous results both on ACM and on PIM and moved here.

Response: The novelty of the paper is highlighted as suggested by the learned reviewer. The subsection has been updated with the results of both ACM and PIM.

Reviewer 3 Report

Please see the attached pdf-file.

Author Response

Thank you very much for providing us an opportunity to revise the manuscript entitled “An Effective Synchronizational Approach to the Stability Analysis for Chaotic Generalized Lotka-Volterra Biological Models Using Active and Parameter Identification Methods” (ID 1600197).

We have endeavored to incorporate all possible revisions/corrections considering the Reviewers’/Editors’ comments, and the changes made in the text are highlighted in Red Colour. The point wise responses are enumerated herein below.

Comment 1:  Equations (1), (2), and (3) define three dynamical systems with the state spaces of the same dimension n. What is the purpose of this constraint? Is it possible to extend the paper's results to the case of distinct state space dimensions?

Response: We have considered three dynamical systems with the state spaces of the same dimension 'n'  in our research work. However, our presented research work can be extended to the case of distinct state space dimensions, for instance, Combination Projective Synchronization in Fractional-order Chaotic Systems with Disturbance and Uncertainty  by Ayub Khan and Uzma Nigar in "International Journal of Applied and Computational  Mathematics", Volume 6, Article Number: 97 (2020).

Comment 2:  Definition 2.1: why do you call systems (1) and (3) chaotic in this definition? Def. 2.1 is a classical synchronization definition and it is not restricted to chaotic systems. Also, no additional assumptions/constraints have been introduce to make (1) and (3) to be chaotic systems. The same comment holds for Def. 2.2 and 2.3.

Response: We have replaced chaotic by master or slave as it was written incorrect by mistake. Further, we have made necessary changes and we have reformulated all Definitions 2.1, 2.2, and 2.3 per the suggestion given by the learned reviewer.

Comment 3: In lines 41-60 (a review of results on chaos synchronization) the output-feedback chaossynchronization can be added (doi: 10.1016/j.ifacol.2019.12.032). Also, besides the completesynchronization it is worth mentioning a less regular behaviors of networks like partial synchronization(doi: 10.1109/TAC.2018.2828780) and multi-clustering (doi:10.1109/TAC.2020.3012528.).

Response: We have cited the references explaining chaos synchronization, behavior of networks like partial synchronization and multi-clustering in literature review part of the paper as suggested by the learned reviewer.

Comment 4: Controller (5) is not rigorously defined. In particular, (i) the arguments of U_i should be added tothe left-hand side. (ii) Also, I cannot find the definition of K_i.

Response: The authors would like to thank the reviewer for this comment. The arguments involving U_i has been added to the left hand side and additionally, the definition of K_i has been added in the revised manuscript as per the suggestion given.

Comment 5: Theorem 1: it is necessary to state that all K_i are positive in order to conclude the required convergence. Also, the proof can be shortened right after (6) since inequality (6) immediatelysuggest that the error dynamics is asymptotically stable provided K_i>0.

Response: According to Lyapunov Stability Theory,  the time-derivative of V should be negative and therefore all K_i are chosen positive. Also, we have reformulated the proof by shortening after Eq. (6) as per the suggestion.

Comment 5:  Eq. (14): the same issue as in comment 4.

Response: The issue has been corrected in the revised manuscript as suggested by the learned reviewer.

Comment 6:  Type setting issue in (32).

Response: The type setting issue in Eq. (32) has been resolved in revised manuscript as per the suggestion given.

Comment 7:  Font size for captions in all figures should be increased.

Response: As suggested,  font size has been increased in all figures in the revised manuscript.

Reviewer 4 Report

The problem studied in this paper is of value in theorem, but the obtained reulsts need to be modified, the active method used in this paper too old to design both simple in form but aslo physical controller, the following references maybe are possible to improve the existing results

[1]Xiaofeng Yi, Rongwei Guo, Yi Qi, Stabilization of chaotic systems with both uncertainty and disturbance by the UDE-based control method,IEEE Access, 8(1): 62471-62477, 2020. 

[2]Rongwei Guo, Yi Qi,Partial Anti-Synchronization in a Class of Chaotic and Hyper-Chaotic Systems. IEEE Access, 9: 46303-46312, 2021.

[3]Rongwei Guo,Projective synchronization of a class of chaotic systems by dynamic feedback control method,Nonlinear Dynamics, 2017, 90(1):53–64.

Author Response

Thank you very much for providing us an opportunity to revise the manuscript entitled “An Effective Synchronizational Approach to the Stability Analysis for Chaotic Generalized Lotka-Volterra Biological Models Using Active and Parameter Identification Methods” (ID 1600197).

We have endeavored to incorporate all possible revisions/corrections considering the Reviewers’/Editors’ comments, and the changes made in the text are highlighted in Red Colour. The point wise responses are enumerated herein below.

Comment 1: Does the introduction provide sufficient background and include all relevant references? It can be improved.

Response: As pointed out by the learned reviewer, we have improved the introduction by adding few newly published researches in literature review part of the paper, for example, references [40], [41] and [42].     

Comment 2: The problem studied in this paper is of value in theorem, but the obtained results need to be modified, the active method used in this paper too old to design both simple in form but also physical controller, the following references maybe are possible to improve the existing results

[1] Xiaofeng Yi, RongweiGuo, Yi Qi, Stabilization of chaotic systems with both uncertainty and disturbance by the UDE-based control method,IEEE Access, 8(1): 62471-62477, 2020. 

[2] RongweiGuo, Yi Qi,Partial Anti-Synchronization in a Class of Chaotic and Hyper-Chaotic Systems. IEEE Access, 9: 46303-46312, 2021.

[3] RongweiGuo,Projective synchronization of a class of chaotic systems by dynamic feedback control method,Nonlinear Dynamics, 2017, 90(1):53–64.

Response: We have cited the references explaining UDE-based control method [52], partial anti-synchronization [25] and projective synchronization [26] in literature review part of the paper as suggested by the learned reviewer.

Round 2

Reviewer 3 Report

The authors have addressed all my comments raised in the previous review round. In my opinion, the paper can be accepted.

Reviewer 4 Report

The revised version is good and suitable for publication in this journal

This manuscript is a resubmission of an earlier submission. The following is a list of the peer review reports and author responses from that submission.

Round 1

Reviewer 1 Report

The authors study in the present paper the projective difference synchronization between identical generalized Lotka-Volterra biological models of integer order by using active control and parameter identification methods.

To do that, they employ Lyapunov stability theory in order to construct the  controllers, with the purpose to ensure the global asymptotical convergence of trajectory following synchronizing errors.

The results are illustrated with numerical simulation in Matlab, showing the accuracy and efficiency of the proposed techniques. Other complementary results and applications are also included.

The results presented in this paper are of interest for a wide range of researchers, and they require the fine manipulation of different techniques. I recommend its publication in Entropy. A couple of comments are also included, for the consideration of the authors:

1-In page 3, paragraph 84 within Definition, I think they mean a vector norm, since, actually, there is no matrix here.

2-Figure 2 in page 5 is much more informative than Figure 1 (p. 4-5), so I would suggest to delete Fig. 1 and keep Fig. 2.

Reviewer 2 Report

In this manuscript, the authors consider three identical Lotka-Volterra biological models and construct the desired controllers which ensure the global asymptotical convergence of trajectory following synchronizing errors. Indeed, the problem studied by the authors is of particular interest. But the presentation of the results causes some questions.

-  Line 120: It is written in the text that "the first master GLV system depicts chaotic behavior". But Fig.1(a) and Fig.2(a) illustrate the phase trajectory that doesn't confirm this. Since the first and second master systems are identical, it is obvious that for both systems, an attractor should be the same. So, for Fig.1(a) and Fig.2(a) it would be better to give a trajectory observed for another initial condition.

- Fig.1(c) and Fig.2(c) demonstrate that the variables of the slave system are negative. Taking into account the physical meaning of the system variables, it seems strange. It should be explained in the text (or other initials should be considered).

-Line 89: "The problem regarding combination synchronization would be converted into traditional chaos control issue if R = S = 0". To avoid misleading, some clarifications\references within this remark are needed.

- Line 92: It is written that " If T = I and R = S = \beta I, then for \beta = 1 it will be reduced to complete synchronization and if \beta = -1 it turns into anti-synchronization". Complete synchronization (within the frame of the cited work of Pecora and Carroll) means the coincidence of the dynamics for all the coupled subsystems. Here, since the control functions are added only to the third (slave) system, this regime is impossible (both master systems are independent). What does "complete synchronization" mean here? It should be clarified. What does "anti-synchronization" mean? It seems to be more appropriate to use the term "anti-phase synchronization", isn't it? To illustrate these regimes for the considered system, the time series for both these cases should be added to the manuscript.

- Finally, the physical meaning of the considered combination difference synchronization error should be discussed.

Reviewer 3 Report

The first part of the paper is well documented and well written. Unfortunately, the scientific soundness and the original contributions are quite small. The authors are careless in the use of notations and, by that, generate inconsistencies and difficulties in reading the paper. For example, the same notation “T” is used for a matrix and for transposition, the quantities q_i and p_i are not clearly defined, neither in (5) nor in (14), the error e is denoted by E in (35), the functions h_i from section 2 become  f_i in section 3, etc.

In fact, section 3, presenting CDS, is practically superfluous as long as CDPS is used in the rest of the paper.

The authors do not explain the form chosen for the controllers in (5). Why this form and what is the advantage? The fact that the expressions (15), (17) and (19) lead to the system (20) is not a solid argument. Moreover, it is quite evident that the synchronization error functions solutions of (20) converge to zero, it is not necessary to check that by numerical investigations. Same remarks work for the computations in section 5.

The investigation methods used by the authors are not new, the model with three identical systems is not a realistic one, the comparative analysis (section 5.2) does not elucidate the novelty and the importance of the results brought by the authors’ approach. In conclusion, I do not recommend the publication.